## Registered report 

psychology

criminal networks, decision-making, disclosure, investigative interviewing, terror groups

**Author for correspondence:**
David A. Neequaye
e-mail: daneequaye@gmail.com

# Exploring how members of illicit networks navigate investigative interviews

David A. Neequaye, Pär Anders Granhag and
Timothy J. Luke

Department of Psychology, University of Gothenburg, Box 500, 40530 Gothenburg, Sweden

DAN, 0000-0002-7355-2784

This study explored how members of an illicit network navigate investigative interviews probing their crimes. We examined how perceived disclosure outcomes, namely, the projected costs and benefits, affect what members choose to reveal. We recruited $N = 22$ groups, maximum of six participants per group. Each group assumed the role of an illicit network and planned for possible interviews with investigators probing into the legitimacy of a business the network owns. All participants underwent an interview after the group planning stage. The results indicated that network members navigate the dilemma interviews bring by disclosing information they perceive would likely yield beneficial (or desirable) rather than costly (or undesirable) outcomes. Additionally, much of the participants' sensitivity to potential costs and benefits was explained by the group of which they are a part: different networks likely respond to costs and benefits in unique ways. This work contributes to understanding how illicit networks manage information disclosure in investigative interviews.

## 1. Introduction

Imagine Detective Doe is investigating an illicit network called MERSA. MERSA is suspected of laundering money via a chain of tanning salons: MERSA's supposed legitimate business. Doe will interview some managers of the tanning salons: the suspected founders of MERSA. The detective is eager to learn from the psychological science of investigative interviewing to assist in preparing for the interviews. Stakeholders advise law enforcement officers to rely on science when soliciting information from human sources (e.g. [1]). This study aims to contribute to efforts like Doe's by examining how members of illicit networks manage information disclosure in an investigative interview. Existing research focuses on the individual strategies of interviewees, even when those interviewees belong to a small group (e.g. [2]). Presently, Detective Doe is unlikely to find information to

understand how illicit networks navigate investigative interviews. This research is an initial attempt to address the challenge of examining the influence of a network's collective planning and the decision-making of individual members therein.

We explored how perceived disclosure outcomes, namely, the projected costs and benefits, affect what networks choose to reveal. The study focused on disclosure pertaining to the network as a whole, not about the individual being interviewed. We focused on a situation where the individual's goals align with their network's goals. This study is *not* about the potential scenario wherein the individual's goals conflict with their network's goals. Examining such a conflict would be useful to the literature, but this aspect is not our current objective.

Our goal is to explore the extent to which group and individual decision-making predict the management of information disclosure about a network. When interviewing someone about their network to what extent does the network the interviewee belongs to predict the *type of information* the interviewee would *choose* to reveal. Are two or more different people from the same network likely to disclose similar kinds of information? Does disclosure in this context resemble individuals independently managing the potential outcomes of revealing information?

Various features could characterize illicit networks, for example, hierarchical differences between members, centralized versus decentralized structures, and the intensity of members' loyalty. To pave the way for future research, our goal at this initial stage is to examine a basic or generic form of a network, in this case, a group consisting of a flat hierarchy wherein the members share a common illicit goal.

# 1.1. Conceptualizing a generalizable taxonomy of information-types

We draw on the disclosure-outcomes management (DOM) model to conceive the types of information an interviewee can disclose [3]. Descriptions of investigative interviews in the field suggest that interviewees typically face a dilemma: conflicting goals wherein some desired outcomes prevent other goals or compete for resources with other goals [4,5]. Consequently, interviewees manage their disclosures such that they reveal some information but not every detail they hold: a finding that features in the published literature [6]. Currently, there is little research examining what interviewees disclose and why they elect to reveal the information they do. Existing research focuses heavily on the amount of information interviewees disclose (see, e.g. [6]). Examining the processes driving those disclosures would be a useful addition to the literature.

The DOM model draws on the established idea that decisions are usually a function of the joint influence of perceived costs and benefits [7–9]. The model predicts that interviewees determine what to disclose via intuitive cost–benefit considerations. Interviewees navigate the conflicting goals of their dilemma by estimating what disclosures would likely yield beneficial (or desirable) rather than costly (or undesirable) outcomes, and they disclose those items accordingly. Thus, an interviewee might be more likely to disclose some information items than others when considering all the entire lot of information the interviewee holds. The DOM model posits that from an interviewee's point of view, the expected outcomes of disclosure, namely, the costs and benefits, can be high or low in magnitude. Such perceived valence depends on two things: the pieces of information the interviewee holds and the interviewee's current dilemma. Using that conceptualization, the model provides a wieldy and generalizable taxonomy of information-types.

## 1.1.1. Low-stakes information

Suppose an interviewee expects no tangible benefits or costs with revealing an information item: the perceived costs and benefits of disclosure are low. These information units have relatively few or unimportant consequences for the interviewee. The DOM model predicts that interviewees would refrain from disclosing low-stakes information to avoid taking unnecessary risks. Such disclosure is not immediately beneficial to navigating the dilemma but carries potential costs.

## 1.1.2. Guarded information

Sometimes, the benefits of revealing a piece of information can be low, while the costs of disclosing it are high. In such cases, interviewees would be unyielding and unlikely to disclose the information. From an interviewee's perspective, the costs of revealing such information far outweigh the benefits.

### 1.1.3. Unguarded information

An interviewee could expect disclosing an information unit to yield a highly beneficial outcome and little to no costly consequences. In this case, revealing the information is in the interviewee's best interests; the interviewee would be maximally likely to disclose things that have the features of unguarded information.

### 1.1.4. High-stakes information

The costs and benefits of some expected outcomes can both be high in magnitude. These situations would elicit a stark motivational conflict. Thus, interviewees are likely to either disclose or withhold the information entirely.

## 2. The present research

### 2.1. Question 1

Preliminary results support the DOM model's conception of information-types [3], but the theory is still nascent. The present research is another attempt to explore the DOM model's tenets, probing how well DOM generalizes to the context of illicit networks. Here we examined the extent to which dilemmas at the level of a network generate the information-types the model predicts.

### 2.2. Question 2

Our next goal was to examine the extent to which individual decision-making and the network a person belongs to independently and jointly predict the kinds of information people disclose about their network. Such knowledge could help practitioners be cognizant about how interviews might go with the various network members under investigation.

The current study aimed to contribute to the literature in three ways. First, we introduced a research design to create mock illicit networks for experiments on investigative interviewing. Then the study tackled two main research questions.

## 3. Method

The hypotheses, power considerations, procedures, materials, data exclusion criteria and analysis plan were preregistered before data collection: https://osf.io/n7ugr.

### 3.1. Participants and design

The research was conducted online via the Zoom video conference platform and Qualtrics. The procedure adhered to the guidelines governing research with human participants, and the IRB of the Federal Bureau of Investigation approved the protocol (Docket No. 629-21). Before commencing the research, participants provided informed consent to the procedure and received a full debriefing after.

The participants were recruited under the guise of a group planning study. They assumed the role of an illicit network, planning for possible interviews with investigators probing into the legitimacy of a business the network owns. We recruited already acquainted participants—for example, friends or co-workers—to serve as a network: typically, network members are not complete strangers. This design choice allowed us to commence test sessions without needing to induce familiarity between participants.

Each network in the present study comprised a maximum of six participants. To our knowledge, there is no absolute number that makes a group or a network, but the literature contains guiding principles. A group should consist of at least three people. Social identity researchers argue that a dyad is not a group: a dyad may not elicit the group dynamics that likely characterize networks, for example, social pressure, coalition formation and deviance from majority decisions [10]. Additionally, we drew on research indicating that computer-mediated communication facilitates an appreciable level of interaction between a group comprising a maximum of six people [11]. We extrapolated that a video conference of six interlocutors would allow participants sufficient opportunity to contribute to group discussions when planning for their potential interviews. Streamlining communication in the

groups was to ensure that all network members understood and were aware of any consensus that emerged during their group planning.

Participants were recruited via a university participant pool and online adverts. We aimed to include a minimum of $N = 20$ networks (six people per network), which would amount to approximately 120 individual participants.

We tested a total of 145 participants to ensure that we achieve our target sample size. Of these prospective participants, 14 were excluded for failing at least one memory check. The final sample included $N = 131$ participants, which amounted to 22 networks. The average age was 25.9 years ($Mdn = 23$, s.d. $= 8.08$). The majority of participants were females 61.10% ($n = 80$), 35.10% males ($n = 46$) and 3.9% preferred to leave their gender unstated ($n = 5$). Also, 66.9% ($n = 88$) of the participants were college educated and 33.1% ($n = 43$) had undergone high school education. Each participant received 200SEK (approx. 24.5USD) as compensation. A test session lasted approximately 40 min.

Each participant made 48 decisions, which provided an approximate total of 6288 observations in the present study. Resource availability guided our choice of sample size[1]. We hope this study will guide future studies to estimate effect sizes and predict how long data collection might take.

## 3.2. Procedure

The appendix contains all the materials reported in the procedure: https://osf.io/rty8q.

### 3.2.1. Phase 1: planning by the entire network

The setting of this phase was a Zoom video conference. An experimenter facilitated the participants using a PowerPoint presentation to explain the protocol of this phase. The appendix contains the experimenter's script.

Each group of participants assumed the role of a network that runs an illegal sports betting business that engages in money laundering. The network fronts as a chain of tanning salons. To enhance group affiliation, each ostensible network commenced the study by determining a name and a slogan for their supposed legitimate chain of tanning salons. We encouraged participants to generate a credible name and slogan to prevent suspicion from law enforcement. For example, Golden Tanning Salons— Get a tan and smile! Creating such fantasy themes and symbolic cues enhances group cohesiveness [12]. After the name and slogan task, the experimenter introduced the remainder of the current phase.

The experimenter informed participants that their network was under suspicion of money laundering. The tax agency had also reported that the group might be under-declaring the income of their supposed legitimate businesses. Hence, police investigators would interview the network members. The group's objective was to extinguish the current suspicions by convincing the investigators that their chain of businesses was legitimate. If the group succeeded in convincing the investigators, the group got to keep their business license. Additionally, the police and tax agency would drop their investigations. If the group failed to convince the investigators, the group might lose their business license; also, the investigators would continue their investigation.

The design aimed to include uncertainty: an illicit network cannot predict, with complete certainty, which members that law enforcement investigators might apprehend and interview. Thus, we informed the group that any of them might be arrested and interviewed in the next phase of the study. The group then received instructions on the nature of the upcoming interviews. In all, the respective interviews would be about three topics on the network. Consequently, each interview consisted of three parts; and all participants would be interviewed on all three topics. Each part would commence with a video presentation in which the interviewer requests information on a topic. For example, the interviewer would ask about how the group started their chain of tanning salons. The network member undergoing the interview would then decide what to disclose on the topic in question. They would execute their decisions by selecting what to disclose from a list of possible information items.

To prepare for the interviews, the group received a background story on the three topics the interviewer would ask about (see appendix). Each topic described the pieces of information to be considered during the group's planning. The experimenter informed the group that during the interview phase, each piece of information would come with two probabilities. The probability that disclosing the information item

---

[1]Our funding for this project allows us to run the present design twice. Thus, we planned for the possibility of conducting a follow-up study if needed. Previous experience in video conference data collection led us to believe we could achieve the current target sample size and a potential follow-up study, given the described constraints.

would contribute to convincing the investigators (presented as 'XX% beneficial'). And the probability that disclosing the information item would jeopardize the likelihood of convincing the investigators (presented as 'XX% dangerous').

The consequences just described were ostensibly linked to participants' compensation using an incentive-compatible procedure. Studies widely use such protocols to elicit true preferences [13]. The group received an initial endowment of 600SEK (approx. 69 USD). We told them that they could double their endowment in the best case, and in the worst case, they could lose the entire endowment. The more the group members disclosed beneficial information during their respective interviews, the more likely the group would increase and possibly double its endowment. The more the group members disclose dangerous information, the likelier they would decrease their endowment and possibly lose it.

We aimed to mimic the semi-cooperative interview scenario wherein the network members are motivated to disclose at least some information (e.g. [14]). Hence, the group was informed that to help dispel suspicion, they must appear to be assisting the investigation by disclosing at least some information. Staying completely silent *might* raise the investigators' suspicions, meaning the group would remain in jeopardy.

The group was told that the plot of the background story would guide them on what might be beneficial or dangerous to disclose. The actual probabilities of beneficial and dangerous disclosures would be revealed during the interview phase. Thus, the group was free to plan how they might tackle the upcoming interviews. Each member's decisions during their potential interview could affect the entire group for better or for worse. In truth, each participant received an equal split of the maximum amount 1200SEK (approx. 138 USD).

After the experimenter presented the instructions the group read the background story, which contained the information the interviewer might ask about. They had a maximum time of 20 min to read and plan and were allowed access to the background story during the interviews. This aspect of the research design aimed to eliminate the possible effects of the pressure to remember verbatim details or forgetfulness.

### 3.2.2. Phase 2: interviewing the network members

After the planning phase, the group received a Qualtrics link to ostensibly take each member to the next phase. The experimenter informed the group that the next phase would determine who might be apprehended for an interview. In truth, each participant was told that the investigators had called them in for an interview. This phase began with assessing the level of affiliation members felt toward their group. We used an adapted version of the Inclusion of Other in the Self (IOS) Scale [15]. That scale is psychologically meaningful and reliably measures interpersonal closeness [16]. We included that scale as an exploratory measure to examine the effect of interpersonal closeness on decisions.

Next, to foreshadow the interview format, we introduced members to the potential outcomes of their disclosures for their group. We reminded members that we would specify the extent to which disclosing each piece of information is likely to be beneficial or costly to their group. The instructions told members that each information unit would come with two probabilities: the probability of a positive outcome (presented as 'XX% beneficial') and the probability of a negative outcome (presented as 'XX% dangerous'). Disclosing a given piece of information would bring a *random* outcome based on the provided probabilities. If the sum of the probabilities of the positive and negative outcomes did not sum to 100%, the remainder would represent neither a positive nor negative outcome.

The instructions told members that their performance would affect their group for better or for worse. Their decisions could boost or diminish their group's average, thereby increasing or decreasing their group's final compensation. This aspect of the design mimicked how a member might affect their network. Each positive outcome would provide an additional 25SEK (approx. 2.8 USD), allowing members to increase their group's initial endowment. Such beneficial disclosures would ostensibly help the group retain their business license and quash the police investigation, which means the group would continue to thrive and make profits. Each negative outcome would detract 25SEK (approx. 2.8 USD), making members reduce their group's initial endowment. Dangerous disclosures would presumably jeopardize the group because they would raise the interviewer's suspicions, meaning the group's business license would likely be revoked, and the group's ability to thrive would dry out. As one can infer, the probability of a negative outcome represented the potential costs of disclosure, and the probability of a positive outcome represented its benefits. We manipulated these probabilities to mirror the DOM models conception of information-types: unguarded (50% beneficial,

15% dangerous), guarded (15% beneficial, 50% dangerous), low-stakes (15% beneficial, 15% dangerous) and high-stakes (50% beneficial, 50% dangerous).

Consistent with the planning phase, the instructions told members that their role included the motivation to appear cooperative by assisting the investigation. Therefore, we told members that staying completely silent may or may not cost their group a random amount of money because silence *might* raise the interviewer's suspicions. That possibility of silence raising suspicions means the group's business license and profits would continue to remain in jeopardy.

The current incentive-compatible protocol aimed to facilitate eliciting the information members were truly willing to disclose. The profitable information, which was *beneficial* to disclose, and the dangerous information, which was *costly* to disclose, was randomly generated. The information-type manipulations suggested the potential outcomes of disclosure. However, members could not determine with complete certainty which disclosures would actually boost or diminish their group's endowment. Hence, there was no way to exploit the process. The protocol demonstrated to participants that the most prudent way to behave was to indicate one's true preferences to take ownership of the decision outcomes. Haphazard responses could not guarantee success or alleviate the risks. Overall, our procedure made the consequences of decisions tangible, not merely imagined.

We included two memory checks (on the incentive compatible protocol) to assist us to flag and exclude those who failed the checks from data analysis (see appendix).

To ensure robustness, we included reminders in Phase 1 and 2 telling participants that communication between group members was prohibited during the interview (see appendix). Before the interview, each participant was required to confirm that they would not communicate with other group members. In addition, we included a check after the interview, asking participants to confirm whether they communicated with any group member during their interview (see appendix). All participants agreed not to communicate with other members during the interviews and they confirmed adhering to the agreement.

## 3.3. The interview

After the instructions, the interview commenced. To better immerse members in their interviewee role and be consistent with the previous instructions, the interviewer spoke to each member via four separate video recordings. An actor portrayed an interviewer: ostensibly, one of the investigators on the case. We recorded the videos using the first-person perspective; the interviewer spoke to the camera as if addressing the viewer directly.

The first video was an introduction in which the interviewer thanked the member for the meeting, described the nature of the subsequent videos and noted that the member had the autonomy to decide what to disclose during the interview. The remaining three videos—respectively—commenced each interviewing block, and the blocks were presented in random order. Thus, the first video allowed a seamless transition to the remaining videos without needing to reintroduce the interviewer every time. In each interviewing block's video, the interviewer directly asked about a topic matching one of the three topics in the background story. See the appendix for the interviewer scripts.

## 3.4. Disclosure decisions

Each interviewer-inquiry was followed by the topic of the background story the interviewer mentioned. And each of those three topics came with a mix of 16 information items—comprising four units of each information-type, presented in random order. Overall, the four respective information-types were presented 12 times; thus, each participant made 48 decisions in total.

Each scenario came with a list of information items. The instructions explicitly told members that they were free to disclose more than one piece of information. They could also disclose nothing if they wished to be silent on the topic in question. After each interviewing block, members received an automated update on their current performance. This feature provided feedback on how the member undergoing the interview was affecting their group. In addition, throughout the interview, participants continually had access to their contribution to the group's average.

The possibility for participants' decisions to increase or diminish their group's endowment aligned with the probabilities describing the information-types. For *unguarded information* (50% beneficial, 15% dangerous), six information items were designated to boost the endowment, two items diminished it, and four items had no effect. *Guarded information* (15% beneficial, 50% dangerous) included two items designated to increase the endowment, six items detracted from it, and four items had no consequence.

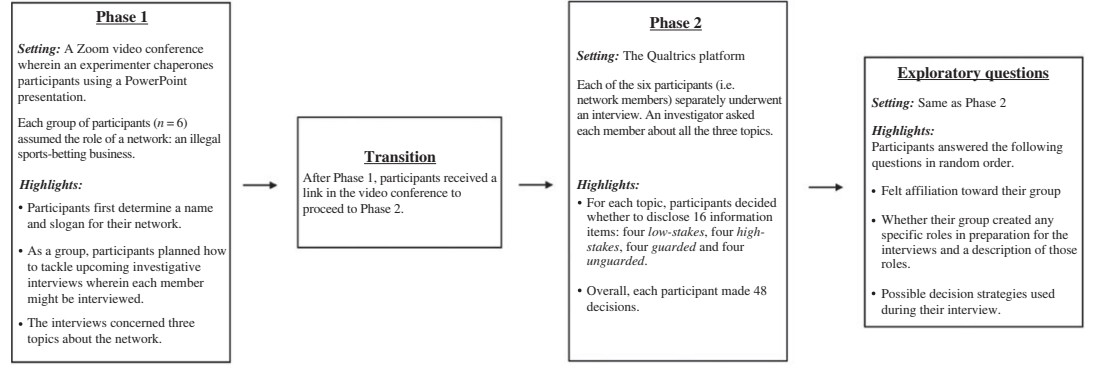

**Figure 1.** Overview of the procedure.

*High-stakes information* (50% beneficial, 50% dangerous) comprised six items designated to earn the group more endowment and six items assigned to lower the endowment. For low-stakes information (15% beneficial, 15% dangerous), we designated two items to increase the endowment, two items decreased it, eight items had no effect. The specific pieces of information that earned, detracted, or had no effect on the group's endowment were randomly generated (see the appendix for the code).

A Qualtrics 'qsf' file is available for readers who want to examine the Phase 2 protocol as participants experienced it: https://tinyurl.com/3ykj84pn.

## 3.5. Exploratory measures

We included exploratory measures to help us generate future research questions (see appendix). Those measures probed the potential strategies and roles the participants devised in preparation for their interviews. Additionally, all participants answered the IOS scale again. We included this IOS measure to examine whether any changes in felt closeness occurred after the interview and explore potential explanations for any observed changes in future studies. Participants received the exploratory measures in random order (figure 1).

## 3.6. Analysis plan

To examine the primary research questions, we fit and compared a series of mixed-effects logistic regression models. The model selection took an additive approach, wherein fixed and random effects were added in progressive steps. We constructed and compared models according to this sequence:

1. A model predicting disclosure decisions (0 = not disclosed, 1 = disclosed) for each piece of information in the interview (48 decisions per participant), with risk level (0 = low, 1 = high) and benefit level (0 = low, 1 = high) as fixed effects, as well as random intercepts for each participant and random intercepts for each topic.
2. A model adding an interaction term for risk and benefit level.
3. A model adding random intercepts for each group (participants nested in groups).
4. A model adding random slopes for each participant for risk and benefit level.
5. A model adding random slopes for each group for risk and benefit level.

Models were compared using likelihood ratio tests (significance threshold = 0.05). We retained for interpretation the model that best fit the data according to these tests (i.e. the latest model in the series that outperforms the previous model). All examined models are documented and reported either in the main text or electronic supplementary material. Models were fit using the *lme4* package [17] for R [18]. Model convergence was evaluated using the *glmer()* function's defaults, but we planned to override the defaults to specify that the optimizer will perform 100 000 function evaluations at maximum. And if a model fails to converge, we planned to remove from consideration for retention and interpretation.

The primary effects of interest were the fixed effects for risk and benefit and the random effects for individual participants and for groups. The risk and benefit effects will provide information about whether the predictions of the DOM model bear out here (research question 1). To support the

hypotheses, the coefficient for benefit should be positive, and the interaction should be negative. The random effects for participants and groups will provide information about the extent to which disclosure decisions are influenced by the individual making the decision and the group to which the individual belongs (research question 2).

To assess statistical power, we conducted a simulation-based power analysis using the *simr* package [19] for R. With a similar design and with a model highly identical to the model we plan to examine, Neequaye *et al.* [3] found a positive coefficient for benefits, $b = 6.33$ 95% CI [4.21, 8.45], and a negative interaction between risk and benefits, $b = -3.76$ 95% CI [−6.64, −0.88]. Using our planned sample size of $N = 120$ participants and using the fixed effects and random effects variances observed in the primary model used by Neequaye *et al.* [3], we examined statistical power for the interaction between risks and benefits under three conditions: (1) with the same effect observed by Neequaye *et al.* [3], (2) with an effect half the size as the previously observed effect, and (3) with an effect equal to the bound of the 95% CI of the original effect that was closer to zero. Under these three assumptions, we found that this sample size will respectively provide 93.50% power for $b = -3.76$, 45.70% power for $b = -1.88$, and 17.00% power for $b = -0.88$. As such, the present study will be well-powered for effects similar in size to the previously observed effects, but it will not have adequate power to detect effects that are substantially smaller. Because of this limitation, if the results are nonsignificant, we will not be able to make claims about the absence of theoretically relevant effects (table 1).

# 4. Results

The data supporting the results and electronic supplementary material can be accessed here: https://osf.io/pq3gh/. The full analysis code is available here: https://github.com/RabbitSnore/DOM-networks.

## 4.1. Group affiliation

We explored the efficacy of our design to assure group affiliation by examining participants' interpersonal closeness ratings. The ratings were stable and high (possible range = 1 to 7). The modal rating before participants underwent their interviews was $Mo = 6$ ($M = 5$, $Mdn = 6$, *s.d.* = 1.66). Ratings were identical after the interview stage $Mo = 6$ ($M = 5.21$, $Mdn = 6$, *s.d.* = 1.64).

We also examined whether the measure of group affiliation taken prior to the interviews might predict participants' decision-making. The findings produced nearly identical results to the models in the primary analyses reported subsequently. The affiliation measure (and its interactions with risk and benefit) had little or no influence on decisions to disclose information. For details on these models, see the electronic supplementary material.

## 4.2. Information disclosure

Across topics, participants disclosed 6.44% of guarded items, 11.38% of high stakes items, 35.98% of low stakes items, and 63.28% of unguarded items. Table 2 contains the descriptives of the proportions disclosed for each information type.

Figure 2 displays the distributions of the total number of information items disclosed by each participant for each type of information. Each information type was represented by 12 items across the three topics, so a participant could have a sum of 0 to 12 disclosures for each information type.

To test our hypotheses, we fit a series of five mixed-effects logistic regression models and compared them using likelihood ratio tests according to our preregistered analysis plan (https://osf.io/n7ugr). Using this approach, we found that the fifth and final model in the series performed best. This model predicted disclosure decisions for each item with the level of risk, benefit and the interaction of those factors as fixed effects. The model also included random intercepts for participants (nested within groups), groups, items, topics and random slopes for risk and benefit level for participants and for groups. The results of that analysis are displayed in table 3. Consistent with predictions, the coefficient for benefit was significant and positive. However, the interaction term for risks and benefits was not significant, and the coefficient for risk was significant and negative.

Additionally, the variance of random effects associated with group membership was considerable for both intercepts and slopes. This large amount of variation suggests that a substantial amount of participants' disclosure decisions and much of their sensitivity to potential risks and benefits is explained by the group of which they are a part.

**Table 1.** Study design template.

| question | hypotheses | sampling plan & test sensitivity rationale | analysis plan | theory that could be shown wrong by outcomes |
|---|---|---|---|---|
| To what extent do self-interest dilemmas at the level of an illicit network generate the information-types the DOM model predicts? | *Low-stakes information:* Interviewees would refrain from disclosing Low-stakes information. *Guarded information:* Interviewees would be unyieldingly unlikely to disclose Guarded information. *Unguarded information:* Interviewees would be maximally likely to disclose things that have the features of unguarded information *High-stakes information:* Interviewees would be likely to either disclose or withhold the information entirely. These four predictions are interconnected and will be tested by the benefit coefficient and the interaction term. | The final sample consisted of 131 participants, comprising 22 networks. Each participant made 48 decisions, which provided a total of 6288 observations in the present study. Resource constraints and the lack of previous research (to estimate an effect size) determined our sample size choice. | A series of mixed-effects logistic regression models (significance threshold = .05). The model selection took an additive approach, wherein fixed and random effects were added in progressive steps. The risk and benefit effects, and their interaction provided information about whether the predictions of the DOM model (i.e. information-types) bear out here. To support the hypotheses, the coefficient for benefit should be positive, and the interaction should be negative. | The DOM model cannot necessarily be disproven here. This research examined the extent to which the model's tenets generalize to illicit networks. As such the study aimed to provide information about the DOM model's generalizability. |

**Table 1.** (*Continued.*)

| question | hypotheses | sampling plan & test sensitivity rationale | analysis plan | theory that could be shown wrong by outcomes |
|---|---|---|---|---|
| To what extent do individual decision-making and the network a person belongs to independently and jointly predict the kinds of information people disclose about their network? | We did not have directional hypotheses concerning this question. Our aim was to provide initial evidence about the strength of the predictors on disclosure decisions. | As above | The same mixed-effects logistic regression models described above. The random effects for participants and groups provided information about the extent to which disclosure decisions are influenced by the individual making the decision and the group to which the individual belongs. | N/A |

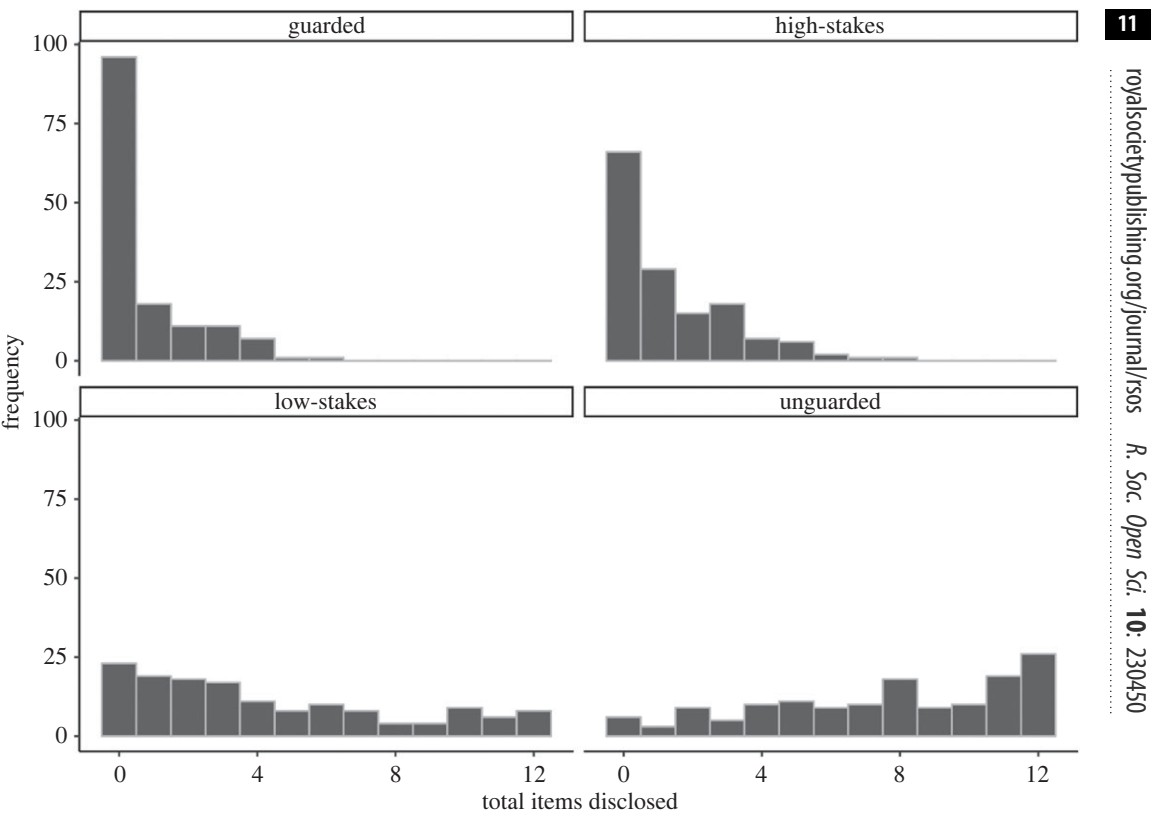

**Figure 2.** Distribution of total number of information items disclosed by each participant, by information type.

**Table 2.** Proportions (standard deviations) of information disclosed, by risk and benefit.

|              | low risk      | high risk     |
|--------------|---------------|---------------|
| low benefit  | .378 (.485)   | .060 (.237)   |
| high benefit | .660 (.474)   | .115 (.319)   |

# 5. Discussion

This research had three objectives: (a) introduce a research design applicable to investigative interviewing experiments on illicit networks; (b) explore how well the DOM model [3] generalizes to the context of illicit networks; and (c) examine the extent to which individual decision-making and the network a person belongs to predict what people choose to disclose.

Overall, the procedure we used worked well to create the mock networks. Recruiting already acquainted participants and inviting them to engage in fantasy themes and create symbolic cues generated strong interpersonal closeness within the respective groups. Such affiliation persisted throughout the study, as demonstrated by the interpersonal closeness ratings after the interview.

The findings also largely supported the DOM model's fundamental tenets. The characteristics of dilemmas influence what interviewees choose to disclose, and they typically share information they perceive would achieve benefits while taking on minimal risks. Participants were the most unyielding with respect to disclosing guarded information. That information-type was disclosed at lower rates than low- and high-stakes information. Participants were singularly forthcoming with unguarded information, which was disclosed at higher rates than low- and high-stakes information.

However, the interaction between risks and benefits was not statistically significant. And the coefficient for risk was significant and negative. Participants disclosed high-stakes information at lower rates than low-stakes information, but not to the same extent as guarded information. In previous findings, participants disclosed high-stakes information at higher rates than low-stakes information, but not to the same extent as unguarded information (i.e. [3]).

**Table 3.** Logistic mixed-effects model results.

| fixed effects | | |
|---|---|---|
| term | *b* [95% CI] | test statistics |
| intercept | −0.99 [−1.60, −0.37] | *z* = 3.15, *p* = .002 |
| risk | −3.17 [−4.04, −2.31] | *z* = 7.18, *p* < .0001 |
| benefit | 1.81 [1.14, 2.48] | *z* = 5.32, *p* < .0001 |
| risk × benefit | −0.41 [−1.19, 0.38] | *z* = 1.01, *p* = .31 |
| random effects | | |
| term | variance (sd) | |
| participants, intercepts | 2.32 (1.53) | |
| participants × risk | 1.68 (1.30) | |
| participants × benefit | 0.74 (0.86) | |
| groups, intercepts | 0.82 (0.90) | |
| groups × risk | 2.07 (1.43) | |
| groups × benefits | 0.88 (0.94) | |
| items, intercepts | 0.38 (0.61) | |
| topics, intercepts | 0.03 (0.17) | |

*Note*: Regression coefficients are log odds ratios.

We cannot identify the specific difference between the study by Neequaye *et al*. [3] and the present study that caused the inconsistency in results; this is an issue for further research. Currently, there are no grounds to claim there was or was not an interaction between risks and benefits. We speculate that the difference in the focus of the dilemma between the studies might be the culprit. In Neequaye *et al*. [3], participants were navigating risks and benefits that had *personal* consequences only. Those participants had complete liberty to take on risks. Conversely, participants' decisions in the present study could affect their *group*. Thus, we suspect that in the current study, participants were less willing to gamble on high risks by disclosing high-stakes information. But they seemed willing to take chances by disclosing low-stakes information, given that sharing such information came with relatively low risks. Taking on low risks for possible benefits in the network's interest is arguably more defensible to one's colleagues than gambling on high risks (e.g. [20]).

Apart from the individual participants' sensitivity to risks and benefits, the results also suggest that knowing the network a prospective interviewee belongs to might assist in predicting their likelihood to disclose information and their sensitivity to risks and benefits of disclosure. There was a substantial amount of variance in disclosure decisions associated with the particular groups to which participants belonged: different networks likely respond to risks and benefits in unique ways. As such, the disclosure tendencies of one member might predict the behaviors of a colleague from the same network, but not a person from another network.

What do the findings mean for our hypothetical investigator? The detective must remember that interviewees are more likely to disclose information if they *perceive* that revealing the specific item under question would benefit their network. Assume the detective is well prepared for the interviews. On the basis of investigations, for example, tips from informants, the detective has an idea of the dilemma MERSA (the hypothetical criminal network) might be contending. Doe can form reasonable predictions about the topics of conversation MERSA members might view as high-stakes information, for example. Suppose Doe discovers that a particular interviewing approach tends to elicit high-stakes information from one network member. That knowledge could be used to plan an interview with another MERSA operative: Doe can now better predict how best to elicit high-stakes information from members of MERSA. Finally, we would remind the detective that the present findings apply to situations where the interviewee's self-interests align with their network's interests, not when a conflict exists. It is crucial to know the dilemma being contended by the interviewee in relation to their network.

## 5.1. Internal and ecological validity

We must address the internal versus ecological validity trade-off, given the interview format of this study. Interviewees select what they want to disclose from a predefined list containing all the relevant information pieces. Typical investigative interviews involve verbal interactions where interviewees self-generate the information items to disclose. Consequently, in verbal interviews, interviewees can waffle, lie or forget about details they would have otherwise disclosed had they remembered. We acknowledge that our study is limited with respect to including the perils of waffling, lying and memory: issues we recommend future research to address. For now, though, we believe our research design is a prudent start, given our objective to examine the mechanisms underlying what network members *choose* to disclose. The present research design allowed participants to choose what to disclose.

Like our procedure, in most studies that have used verbal interviews, participants assume interviewee roles via background stories (e.g. [21]). Those stories guide the coding of verbal interviews by providing predefined criteria of what constitutes true or legitimate disclosures instead of outlandish and false ones. We acknowledge that coding verbal interviews can generate new information items that researchers did not predict from their background stories, but this aspect was not the focus of the present study. Moreover, coding essentially whittles down verbal interviews into a list of legitimate items interviewees have disclosed. Our procedure retains the essential aspect of flagging legitimate disclosures and eliminates potential coding errors.

Our justifications are not to dismiss the psychological realism verbal interviews can bring. Indeed, the quantitative feedback our protocol provided is far more definitive and informative than what might manifest in an actual interview. We agree and argue that our design addresses the essential research question. To what extent do perceived outcomes influence what network members choose to disclose? We deliberately ensured internal validity, given this early stage of examining the mechanisms of what network members *choose* to disclose.

That notwithstanding, the critic is well within their rights to have the following concern. Did our procedure overly exclude nuance? For example, an interviewer's verbal and nonverbal reactions might affect interviewees' appraisal of potential disclosure outcomes. In this study, participants received the probabilities of disclosure outcomes before disclosure and the consequences of decisions afterward. These aspects of our design generally aimed to mimic desirable and undesirable outcomes, including the perceived positive and negative interviewer reactions. Note that the present research design made the consequences of participants' decisions tangible not merely hypothetical. Future research can build on the foundation our procedure provides. Such future work can examine whether interviewees flag less definitive potential disclosure outcomes and the corresponding effect on disclosure.

## 6. Concluding remarks

This research provides initial evidence that the DOM model's core tenets might be applicable to the context of illicit networks. But more research is required to arrive at firmer conclusions about the similarities and differences between questioning independent individuals and members of groups. It seems network members navigate the dilemma interviews bring by disclosing information they perceive would likely yield beneficial (or desirable) rather than costly (or undesirable) outcomes. We hope this work will inspire more research on illicit networks with a focus on why members therein choose to disclose the information they reveal.

## Author note

This research is a Registered Report reviewed and recommended by PCI Registered Report.

Link to Stage 1 Recommendation: https://rr.peercommunityin.org/articles/rec?id=154
Link to Stage 2 Recommendation: https://rr.peercommunityin.org/articles/rec?id=256

**Data accessibility.** All data supporting the findings in this research are publicly available on the open science framework repository (osf.io).

**Authors' contributions.** D.A.N.: conceptualization, data curation, formal analysis, funding acquisition, investigation, methodology, project administration, writing—original draft, writing—review and editing; P.A.G.: funding acquisition, methodology, writing—review and editing; T.L.: conceptualization, data curation, formal analysis, visualization, writing—review and editing.

All authors gave final approval for publication and agreed to be held accountable for the work performed therein.

**Conflict of interest declaration.** The authors of this article declare that they have no financial conflict of interest with the content of this article.

**Funding.** This research is funded by the United States High-Value Detainee Interrogation Group Contract 15F06720C0002022 awarded to David A. Neequaye and the University of Gothenburg. Statements of fact, opinion, and analysis in this work are those of the authors and do not reflect the official policy or position of the Federal Bureau of Investigation or the United States Government.

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
