## [Peer Review File · Royal Society Open Science]

Review History

Decision letter (RSOS-230450.R0)

Dear Dr Neequaye

On behalf of the Editor, I am pleased to inform you that your Manuscript RSOS-230450 entitled "Exploring How Members of Illicit Networks Navigate Investigative Interviews" has been accepted in principle for publication in Royal Society Open Science.

Please note the RSOS editorial office will now complete the Stage 2 submission on your behalf.

In addition, please can you kindly provide us with a short 100 word media summary for the paper?

on behalf of Professor Chris Chambers (Associate Editor) and Chris Chambers (Registered Reports Editor, Royal Society Open Science)
openscience@royalsociety.org

Associate Editor Comments to Author (Professor Chris Chambers):

Associate Editor: 1

Comments to the Author:

(There are no comments.)

Associate Editor: 2

Comments to the Author:

(There are no comments.)

Author's Response to Decision Letter for (RSOS-230450.R0)

See Appendix A.

Decision letter (RSOS-230450.R1)

Dear Dr Neequaye:

I am pleased to inform you that your manuscript entitled "Exploring How Members of Illicit Networks Navigate Investigative Interviews" is now accepted for publication in Royal Society Open Science.

Please remember to make any data sets or code libraries 'live' prior to publication, and update any links as needed when you receive a proof to check - for instance, from a private 'for review' URL to a publicly accessible 'for publication' URL. It is also good practice to add data sets, code and other digital materials to your reference list.

Royal Society Open Science is a fully open access journal. A payment may be due before your article is published. Please note that, if the corresponding author of your paper is based at an institution covered by one of our Transformative Agreement deals, your fees may be covered by the deal – please check the list of eligible institutions at <https://royalsociety.org/journals/authors/read-and-publish/read-publish-agreements/>. The Royal Society has partnered with Copyright Clearance Center's (CCC's) RightsLink service to allow authors to pay article processing charges or page charges. After your manuscript has been accepted, the corresponding author will receive an email from CCC with the subject "Please submit your article processing/open access charge(s)/page charges" inviting you to pay your charges or request an invoice. The email from CCC will come from the email domain @copyright.com (if you have any queries regarding fees, please see <https://royalsocietypublishing.org/rsos/charges> or contact authorfees@royalsociety.org). If you request an invoice, it will be sent to you from CCC. It is important to be cautious about payment scams. **If you receive an email or text message requesting payment and have any concerns, we recommend contacting us through our website, rather than clicking on any links. The Royal Society will never ask you to make a direct payment.**

The proof of your paper will be available for review using the Royal Society online proofing system and you will receive details of how to access this in the near future from our production

office (openscience_proofs@royalsociety.org). We aim to maintain rapid times to publication after acceptance of your manuscript and we would ask you to please contact both the production office and editorial office if you are likely to be away from e-mail contact to minimise delays to publication. If you are going to be away, please nominate a co-author (if available) to manage the proofing process, and ensure they are copied into your email to the journal.

Your feedback matters - please spend 5 minutes leaving anonymous feedback about your experience of Registered Reports at this journal, as an author or reviewer:
https://registeredreports.cardiff.ac.uk/feedback/feedback/decision_letter.php

This feedback is collected by the Registered Reports Community Feedback website, which is an independent service and research project, being undertaken by Cardiff University.

Follow Royal Society Publishing on Twitter: @RSocPublishing
Follow Royal Society Publishing on Facebook:
<https://www.facebook.com/RoyalSocietyPublishing/>
Read Royal Society Publishing's blog:
<https://royalsociety.org/blog/blogsearchpage/?category=Publishing>

Appendix A

This research is a Registered Report reviewed and recommended by PCI Registered Report.

- Link to Stage 1 IPA (PCI RR): <https://rr.peercommunityin.org/articles/rec?id=154>
- Link to Stage 2 Recommendation: <https://rr.peercommunityin.org/articles/rec?id=256>